# Mathematical Modeling of Hydrodynamics in Bioreactor by Means of CFD-Based Compartment Model

**Agnieszka Krychowska** [1], **Marian Kordas** [1], **Maciej Konopacki** [1], **Bartłomiej Grygorcewicz** [2], **Daniel Musik** [1,3], **Krzysztof Wójcik** [1,3], **Magdalena Jędrzejczak-Silicka** [4] **and Rafał Rakoczy** [1,*]

[1] Faculty of Chemical Technology and Engineering, West Pomeranian University of Technology in Szczecin, Piastów Avenue 42, 71-065 Szczecin, Poland; agnieszka.krychowska@gmail.com (A.K.); marian.kordas@zut.edu.pl (M.K.); maciej.konopacki@zut.edu.pl (M.K.); daniel@escglobal.co.uk (D.M.); krzysztof@escglobal.co.uk (K.W.)

[2] Department of Laboratory Medicine, Pomeranian Medical University in Szczecin, al. Powstańców Wielkopolskich 72, 70-111 Szczecin, Poland; bartlomiej.grygorcewicz@pum.edu.pl

[3] ESC Global Sp. z o.o., Słoneczny Sad 4F, 72-002 Dołuje, Poland

[4] Faculty of Biotechnology and Animal Husbandry, West Pomeranian University of Technology in Szczecin, Janickiego Street 32, 71-270 Szczecin, Poland; magdalena.jedrzejczak@zut.edu.pl

\* Correspondence: rafal.rakoczy@zut.edu.pl

**Abstract:** This study presents the procedure of deriving a compartmental model (CM) based on an analysis obtained from the computational fluid dynamics (CFD) model of a bioreactor. The CM is composed of two parts, a structural (that takes into account the architecture of the mathematical model), and a parametric part (which contains the extrinsic parameters of the model). The CM is composed of the branches containing the set of perfectly mixed continuous stirred-tank reactors (CSTRs) in a configuration that matches the bioreactor's flow patterns. Therefore, this work's main objective was to develop a mathematical model that incorporated the flow field obtained by CFD technique. The proposed mathematical model was validated by means of the experimental data in the form of the residence time distribution (RTD) measurements.

**Keywords:** biochemical engineering; bioreactors; mathematical modeling; modeling

## 1. Introduction

The bioreactor is defined as facilities that enable the efficient operation of microbiological processes by controlling culture parameters and managing its optimal conditions, simultaneously limiting the possibilities of its contamination [1,2]. They can also be defined as an engineered device designed for optimal growth through a biocatalyst and microorganisms' metabolic activity. The bioreactors mainly present optimal conditions for the microorganisms' cultivation. These conditions could be modified to trigger microorganisms' metabolic activity under given conditions [3,4]. Bioprocessing's biggest problem is understanding and modeling the biological particle's hydrodynamics [5,6]. The analysis of hydrodynamics in bioreactors may be carried out by means of the "systemic modelling" proposed by Levenspiel [7]. According to Levenspiel (2002), the use of the plug-flow reactor (PFR) with an axial dispersion model or the use of the continuous stirred-tank reactors (CSTRs) in a series model can be used for the bioreactor liquid hydrodynamics modeling [8]. The initial structure of the mathematical model is derived from the tracer experiments' interpretation [8].

Danckwerts introduced the concept of residence time distribution (RTD) to identify situations in non-ideal mixing systems [9]. Due to that, the input of a non-reactive tracer into a mixing system might provide descriptive information on the hydrodynamics and transfer processes [10]. The pulse–response curve analysis might provide information about the mixing process [11]. The non-ideal mixing zones inside the flow (e.g., dead zones, by-passing paths) may be identified by analyzing the shape of experimental curves [12]. Additionally, this technique allowed to establish the flow patterns in the mixing systems and can be considered as a quantitative method for defining the behavior of mixing devices [13].

Necessary information concerning the mixing process can be obtained by the utilization of the Navier–Stokes equations [14]. The development of computational fluid dynamics (CFD) provides qualitative and quantitative information concerning the hydrodynamics and mixing performance in various types of mixing systems [15]. It should be noted that this technique may be applied to obtain the detailed hydrodynamic on meso- and macro-mixing levels [14]. This approach is based on the mixer or reactor description by a CFD-based structural and functional network of compartments. The obtained flow fields may be used to define a compartmental model (CM), and the exchange flows between the selected arbitrarily compartments [16]. The CM may be successfully applied for the prediction of hydrodynamics in the analyzed systems [17,18]. The hybrid approach based on the combination of CFD and CM techniques is also discussed in the literature [19,20]. Hristov et al. (2004) reported that the CFD-based compartmental model could be used to describe the performance of novel impeller geometry in a triple impeller bioreactor [21].

The current study's main aim was to develop a CFD simulation-based method for defining a mathematical model describing the commercial bioreactor (BioFlo® 415). Briefly, a methodology to develop the CM based on the CFD provided hydrodynamic information. The application of the obtained flow fields from numerical simulations allowed to compose the models' architecture with the branches contacting the system of perfectly mixed CSTRs. Based on the CFD and compartmental modeling approach, the hydrodynamic of the tested mixing system may be described using the system of differential equations characterizing the mass transfer balances for the proposed mathematical model compartmental structure. Moreover, the RTD experiment was performed to compare the tested bioreactor's obtained curve to the CM. In this case, the RTD-based response curves may be used as the qualitative comparison with the proposed model predictions.

## 2. Materials and Methods

### 2.1. Experimental Set-Up

The experiments were carried out with the use of the laboratory-scale stirred dual–Rushton bioreactor BioFlo® 415 system (Eppendorf, Enfield, CT, USA). A schematic, design and the main geometrical parameters of the BioFlo® 415 apparatus are presented in Figure 1.

The experimental measurements of the mixing process for the BioFlo® 415 were performed using a cylindrical glass vessel mounted in the metal housing with a height of the liquid to vessel diameter ratio equal to 2.31 (HL = 345 mm; D = 149.5 mm; volume V = 7.0 dm$^3$). Tap water at a temperature ranging from 20 °C and 25 °C was used as a working liquid. Volumetric flow rates of the working liquid varied from 10 to 60 dm3·h$^{-1}$. Measurements were performed for the various impeller speeds (100–600 RPM). Moreover, the untested bioreactor's mixing behavior with no impeller used was also analyzed (0 RPM).

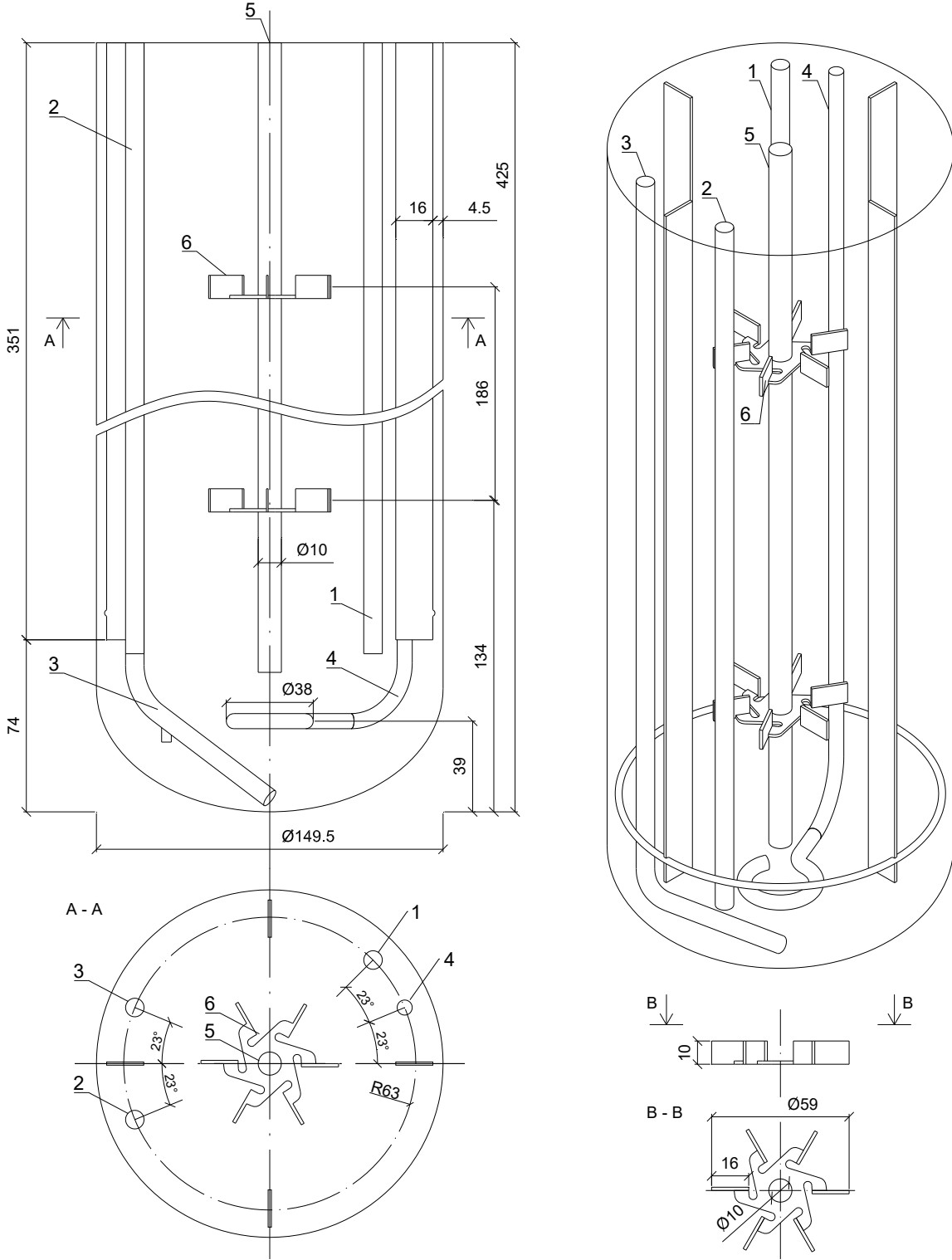

**Figure 1.** Sketch of the BioFlo 415 bioreactor: 1—tube for the temperature sensor; 2—sampling tube; 3—batching tube; 4—sparger; 5—shaft; 6—impeller.

### 2.2. Tracer Experiments

The mixing process occurring in the BioFlo® 415 was evaluated by the utilization of the stimulus–response technique to generate RTD measurements. As the tracer, a saturated NaCl solution (25%, *w/v*) was used. In the current study, the RTD was analyzed by instantaneously injecting

a tracer (a pulse input) into the flow system by inlet and measuring the conductivity at the outlet as a function of time. The BioFlow®415 connected with the tracer batching system is presented in Figure 2.

a)                                                                                          b)

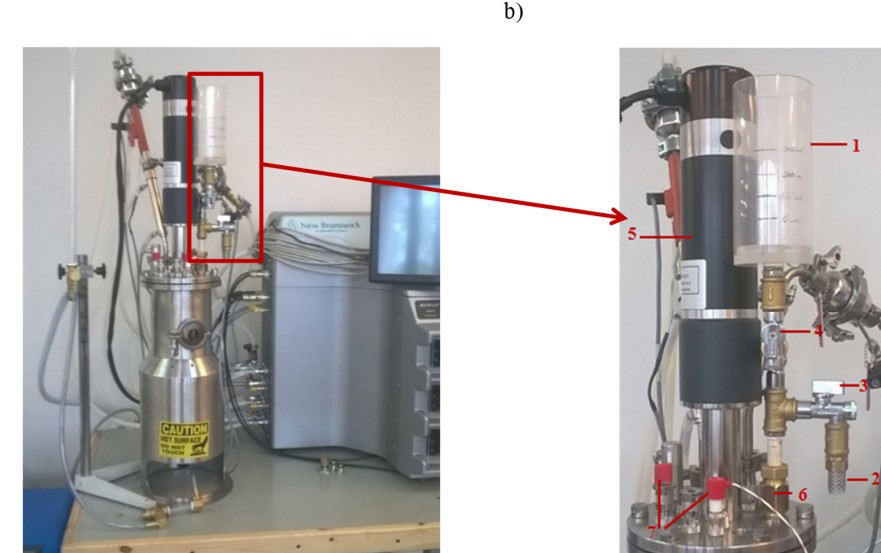

**Figure 2.** The view of the bioreactor BioFlo®415 (**a**) with the batching system of the tracer (**b**). The batching system of the tracer consisted of 1—the batcher; 2—the water supply hose to the mixing system; 3—the cut-off valve for the water supply hose to the mixing system; 4—the dosage valve for the tracer; 5—the power drive of the bioreactor; 6—the connector pipe for the water supply to the mixing system; and 7—the liquid level indicator in the mixing system.

For each measurement, 100 mL tracer was infused into the liquid surface near the bioreactor wall. The Dirac delta function (δ-Dirac function, Equation (1)) describes the applied impulse:

$$\delta(t - \tau_0)\begin{cases} = 0 \text{ for } t \neq \tau_0 \\ \neq 0 \text{ for } t = \tau_0 \end{cases} \text{ where } \int_{-\infty}^{+\infty} \delta(t - \tau_0)\, dt = 1 \tag{1}$$

Two conductivity probes measured the temporal changes in the NaCl concentrations. Samples were collected in 1 s intervals until the disappearance of the tracer in the tested volume. The standard patterns of liquid–tracer conductivity measurements are presented in Figure 3.

a)                                                                                          b)

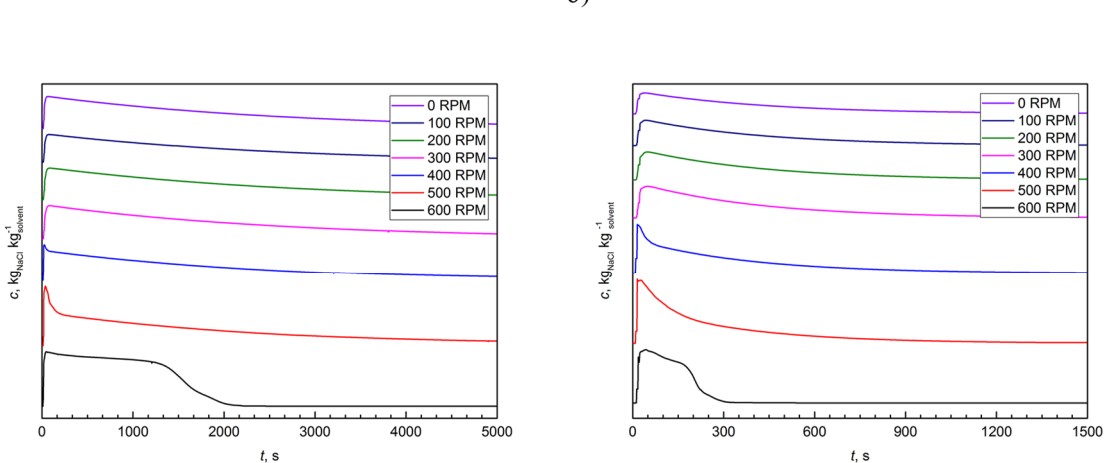

**Figure 3.** The typical example of residence time distribution (RTD) curves at different flow rates: (**a**) 10 dm$^3$·h$^{-1}$; and (**b**) 60 dm$^3$·h$^{-1}$.

### 2.3. CFD Simulations

The flow patterns represent fluid flowing in a stirred vessel and may be generated by using the CFD technique. The numerical simulations using CFD codes present qualitative and quantitative information about the performance and hydrodynamics of the agitated systems' mixing. The usage of numerical methods permitted to build out spatially and temporally representative differential equations based on distributed models of the parameter for the tested bioreactor systems. The CFD's capability to simulate the flow distribution numerically in a mixer can contribute significantly to the comprehension of the mixing processes. The simulation also enables time-saving, is more cost-effective, and improves design optimization [22]. In the current study, the CFD method was used to analyze the BioFlo®415 bioreactor systems' mixing process and to evaluate the velocity profiles and flow patterns. A CM might develop the influence of mixing on a stirred-tank bioreactor prediction performance with the incorporated CFD-simulated flow fields [16].

The ANSYS Workbench 14.5, a commercial CFD-based package, was used for numerical computations. The BioFlo®415 bioreactors' geometry was created with the use of the AutoCAD software to evaluate the liquid flows in the tested system. The same computational model of geometries, consisting of 1.9 million tetrahedral volume elements, for the tested BioFlo®415 bioreactor was created. The CFD mesh was generated in ANSYS Meshing software based on imported design Modeler geometries. The ANSYS CFX software was used to obtain numerical computations of the flow patterns. The control volume formulation was used for solving the mass and momentum equations. The Reynolds-averaged Navier–Stokes (RANS) equations with the k-ω turbulence-closure model were solved by the three-dimensional finite volume CFD-code. The two-equations k-ω based model proposed by Wilcox (1988) allows computing complex laminar and turbulent flows in static mixers [23]. This study model also contains modifications for low-Reynolds number effects, compressibility, and shear flow spreading. The simulation of bioreactor agitators was performed by means of the multiple reference frame (MRF) approach. This is a steady-state approach (which simplifies the solution), where the volume inside the bioreactor can be approximately described by two zones—stationary and rotational (around the impeller). Inside the rotating zone, the fluid flows around the impeller with the corresponding velocity instead of the impeller's physical motion. The variables calculated for the rotating zone are then translated to the adjacent zone through the interface, allowing them to calculate the boundary's fluxes. It is a commonly used approach for simulations of the turbomachinery, e.g., turbines or mixers [24].

The typical computational velocity profiles for the tested bioreactor are presented in Figure 4. This figure compares the computed flow field for the selected operational conditions (10 and 60 $dm^3 \cdot h^{-1}$; $N = 100$ and 600 RPM). To facilitate the analysis of the three-dimensional hydrodynamics, the velocity was shown on two-dimensional planes. When the residuals for the equations of continuity and momentum were below $10^{-7}$, simulations were assumed to converge.

The liquid flow patterns created by two Rushton turbines are independent of each other when the impeller clearance is greater than the impeller diameter [25]. In the current study, the ratios of the impeller spacing and the impeller diameter were equal to 3.15. Therefore, four stable ring vortices were formed because the turbine gave its characteristic upper and lower ring vortices. This flow type may be characterized as a parallel flow pattern. Each impeller generates a radial jet outward and divides into near-wall streams. Additionally, the turbulent regions characterized by high shear forces and rapid mixing were generated near the rotating impellers. It should be noted that the knowledge of the regions with the strong mixing effect inside the tested bioreactor may be essential for mathematical modeling using compartmental modeling.

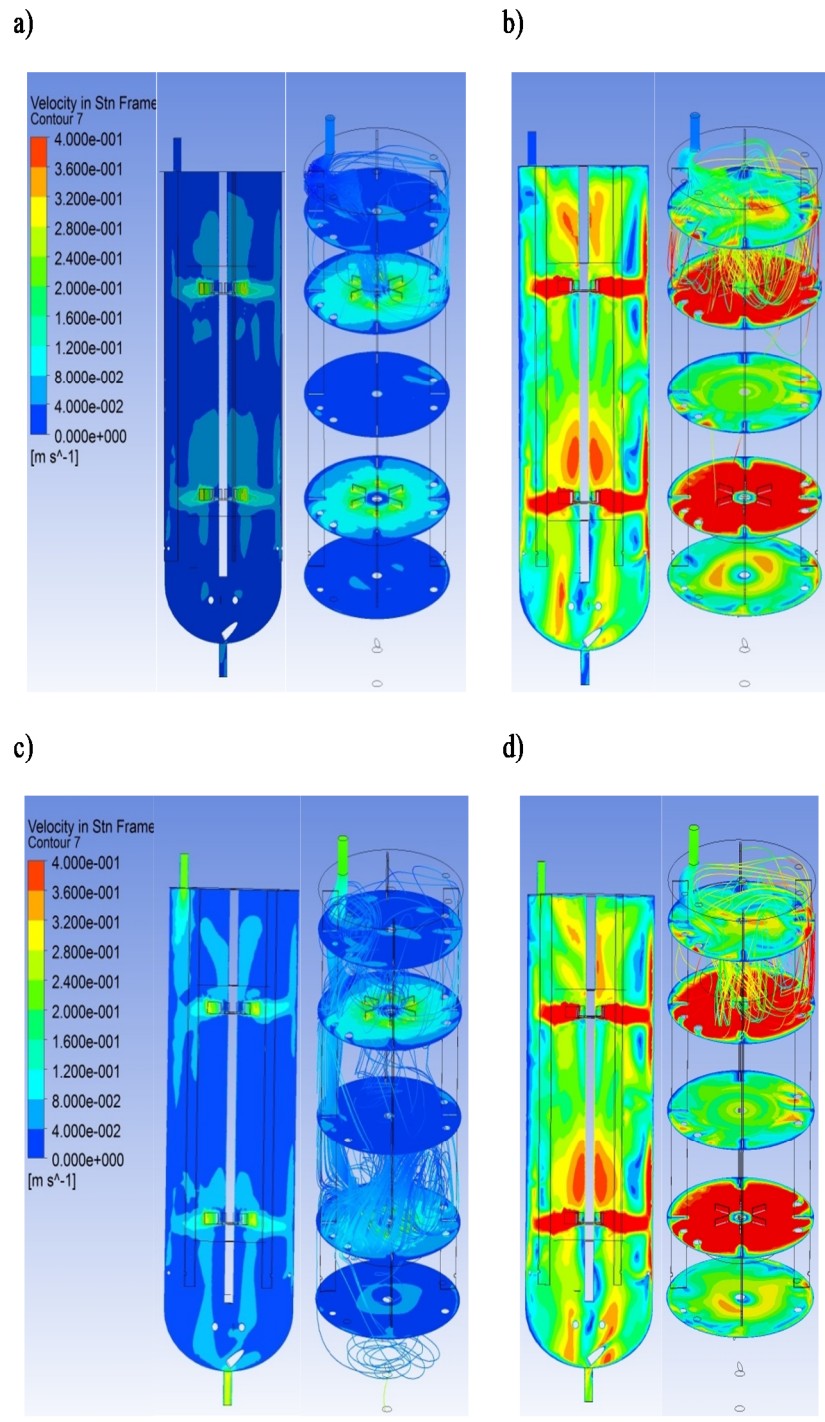

**Figure 4.** The typical example of the velocity field and overall flow pattern for the tested bioreactor in vertical and horizontal planes: (**a**) $\dot{V}$ = 10 dm$^3$·h$^{-1}$ and $N$ = 100 RPM; (**b**) $\dot{V}$ = 10 dm$^3$·h$^{-1}$ and $N$ = 600 RPM; (**c**) $\dot{V}$ = 60 dm$^3$·h$^{-1}$ and $N$ = 100 RPM; (**d**) $\dot{V}$ = 60 dm$^3$·h$^{-1}$ and $N$ = 600 RPM.

## 3. Result and Discussion

The bioreactor mixing process might be characterized by the compartment model (CM). This model allowed to predict the effect of mixing in the bioreactor and to capture the essential features of macro-mixing in the tested stirred-tank reactor. The CM is composed of two parts, one structural (that takes into account the architecture of the mathematical model), and a parametric part (contains the extrinsic parameters of the model) [26]. Generally, the CM is based on the bioreactor's division

into several compartments, and the generated compartments used to estimate the mixing process in bioreactors are selected arbitrarily [16]. Therefore, the CFD was highlighted as an emerging alternative for the explanation of the mixing problems. The CFD techniques numerically simulate the flow in a mixing system and provide a better description of hydrodynamics in the analyzed mixer [22]. The CFD simulation-based results might be valuable for predicting the mixing in bioreactors with relatively good accuracy [27]. Moreover, the CFD simulations may be used to develop the structure of the CM. This approach is called the CFD-based compartment model (CFD-CM) [5].

The present study attempted to develop the CFD-CM, in which the CFD results of the velocity field in the tested bioreactor (BioFlo®415) provides the input for the mathematical modeling in the form of CM. This modeling characterizes the bioreactor as a spatially localized functional compartments network. The connectivity, number, and volume of compartments were determined by using a detailed analysis of the CFD results. This concept is illustrated in Figure 5, where five subdomains of the typical CFD results were mapped into a compartmental model. The proposed approach splits the entire bioreactor into a connected, well mixed compartment, as shown in Figure 5. The connections between the individual compartments were proposed based on the obtained velocity vectors, velocity contours, and flow direction-based numerical simulations [16]. In the presented approach, the compartments should be understood as a mass of well mixed, homogenous liquid that behaves uniformly.

The proposed compartment model is visualized by a diagram wherein the rectangles present compartments, and the arrows symbolize the liquid exchange. This approach might be efficiently used to model the mixing process in reactors to incorporate micro-mixing effects [28]. Correa (1993) showed that this idea might be applied to describe turbulence in chemical reactions [29]. Fan et al. (1970) demonstrated that compartmental modeling was used to describe powder mixing [30]. It should be noted that this method can be applied to any mixing process only if there is information characterizing the number of compartments needed. The number of compartments might be obtained experimentally with the use of the RTD measurements [13]. Furthermore, the mixing process's prediction with the use of the CFD technique can provide detailed hydrodynamics modeling [31].

According to the obtained results, it was evident that the real mixing pattern of the proposed CM can be described as the system of perfectly mixed CSTRs. The final structure of the CM may be defined employing the differential equations (Equation (2)):

$$\begin{cases} V_1\rho\frac{dC_1(t)}{dt} = \dot{q}C_0 + \dot{q}\gamma C_2 - \dot{q}(1+\gamma)C_1 \\ V_2\rho\frac{dC_2(t)}{dt} = \dot{q}(\alpha+\gamma)C_1 + \dot{q}\gamma C_3 - \dot{q}(\alpha+2\gamma)C_2 \\ V_3\rho\frac{dC_3(t)}{dt} = \dot{q}(\alpha+\gamma)C_2 + \dot{q}\gamma C_4 - \dot{q}(\alpha+2\gamma)C_3 \\ V_4\rho\frac{dC_4(t)}{dt} = \dot{q}(\alpha+\gamma)C_3 + \dot{q}\beta C_1 + \dot{q}\gamma C_5 - \dot{q}(\alpha+\beta+2\gamma)C_4 \\ V_5\rho\frac{dC_5(t)}{dt} = \dot{q}(1-\alpha-\beta)C_1 + \dot{q}(\alpha+\beta+\gamma)C_4 - \dot{q}(1+\gamma)C_5 \end{cases} \quad (2)$$

where: $c_i$—tracer concentration, $kg_{NaCl}\cdot(kg_{solvent})$-1; $\dot{q}$—mass flow rate, $kg\cdot s^{-1}$; $t$—time, s; $V_i$—compartment volume, m3; $\alpha$, $\beta$, $\gamma$—parameters of the mathematical model. For the calculation of the system of the differential equation (Equation (2)), the compartment volumes ($V1$–$V5$), and the flow rates ($\dot{q}$) must be known. It should be noted that the compartmental structure is based on the CFD-generated simulations of the hydrodynamic. Therefore, the compartment volumes may be estimated, considering the overall flow pattern [32].

The proposed mathematical model of the mixing process in the BioFlo®415, consisting of Equation (2), was solved using the Matlab/Simulink software. Figure 6 shows an overview of the block model built in the Simulink based on the CM and the set of differential equations presented above.

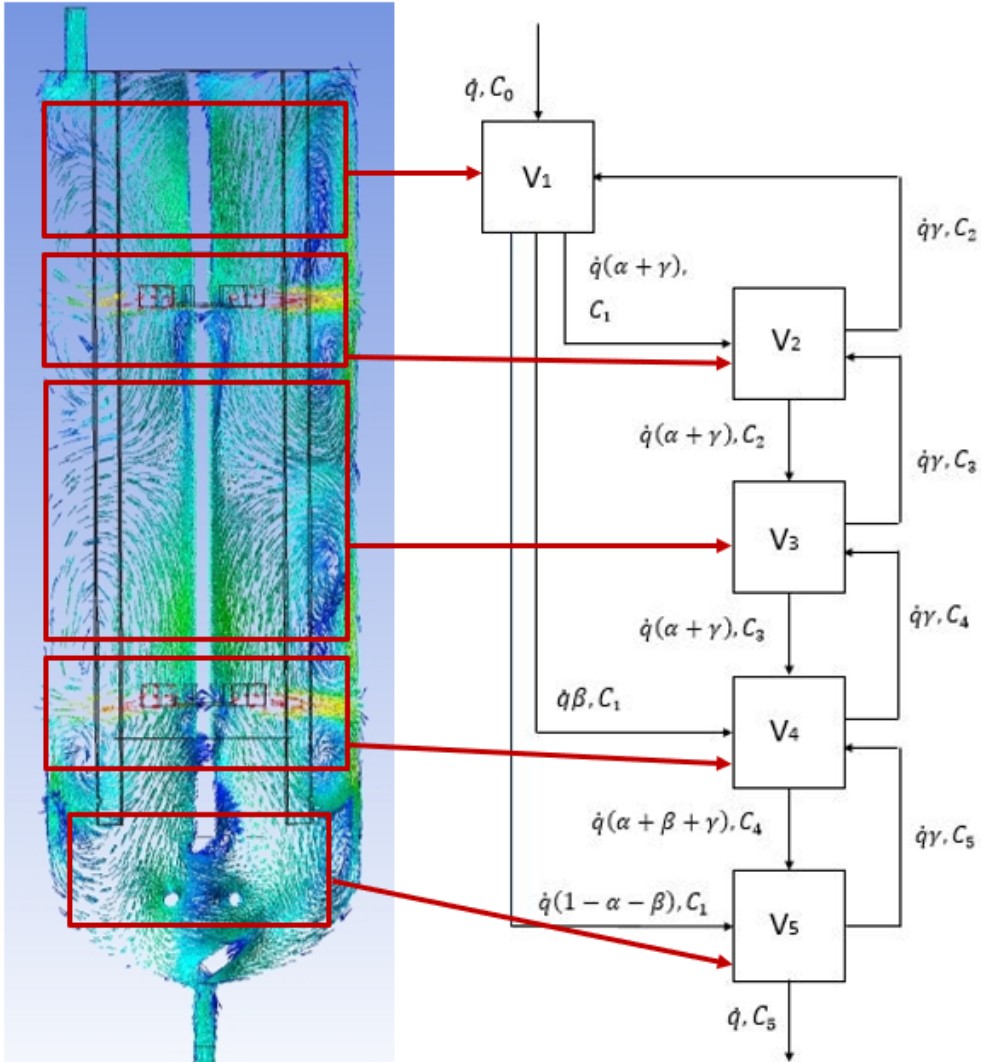

**Figure 5.** Mapping of the CFD results (for $\overset{\bullet}{V}$ = 30 dm$^3$·h$^{-1}$ and $N$ = 300 RPM) into a comaprtmental model ($V_1$—inlet subdomain; $V_2$—upper turbine subdomain; $V_3$—intensive mixing subdomain; $V_4$—bottom turbine subdomain; $V_5$—outlet subdomain). The parameters of the mathematical model: $\alpha$, $\beta$, $\gamma$; volumetric flow rate: 10–60 dm$^3$·s$^{-1}$, compartment volume: $V_1$ = 0.1 dm$^3$; $V_2$ = 2.2 dm$^3$; $V_3$ = 1.6 dm$^3$; $V_4$ = 2.2 dm$^3$; $V_5$ = 0.9 dm$^3$.

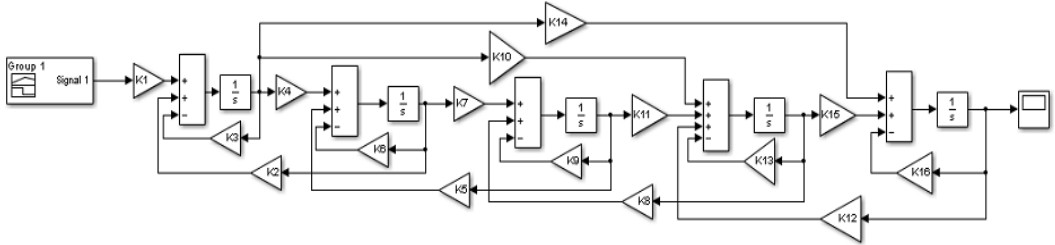

**Figure 6.** The mathematical model in the Matlab/SimulinkTM environment (where: $K1 = \frac{\dot{q}}{V_1\rho}$, $K2 = \frac{\dot{q}}{V_1\rho}\gamma$, $K3 = \frac{\dot{q}}{V_1\rho}(1+\gamma)$, $K4 = \frac{\dot{q}}{V_2\rho}(\alpha+\gamma)$, $K5 = \frac{\dot{q}}{V_2\rho}\gamma$, $K6 = \frac{\dot{q}}{V_2\rho}(\alpha+2\gamma)$, $K7 = \frac{\dot{q}}{V_3\rho}(\alpha+\gamma)$, $K8 = \frac{\dot{q}}{V_3\rho}\gamma$, $K9 = \frac{\dot{q}}{V_3\rho}(\alpha+2\gamma)$, $K10 = \frac{\dot{q}}{V_4\rho}(\alpha+\gamma)$, $K11 = \frac{\dot{q}}{V_4\rho}\beta$, $K12 = \frac{\dot{q}}{V_4\rho}\gamma$, $K13 = \frac{\dot{q}}{V_4\rho}(\alpha+\beta+2\gamma)$, $K14 = \frac{\dot{q}}{V_5\rho}(1-\alpha-\beta)$, $K15 = \frac{\dot{q}}{V_5\rho}(\alpha+\beta+\gamma)$, $K16 = \frac{\dot{q}}{V_5\rho}(1+\gamma)$).

The model of differential equations was integrated numerically by using the Runge–Kutta method. The quantification of the fit goodness was carried out by the sum of the squares of deviation minimalization, $\varepsilon$, among the quantified and the expected RTD curves. The objective function minimized here is given by the following equation (Equation (3)):

$$\varepsilon = \frac{1}{N} \sum_i \left[ c(t_i)|_{experimnet} - c(t_i)|_{model} \right]^2 \tag{3}$$

Figure 7 shows the typical comparison of the CM-based RTD curves versus the investigational data. The curves demonstrate good agreement for all of the tested hydrodynamic conditions. This suggests that the current study's analyses enable obtaining a good quality prediction for the RTD of the tested bioreactor and representative information about the BioFlo®15mixing properties (Figure 4).

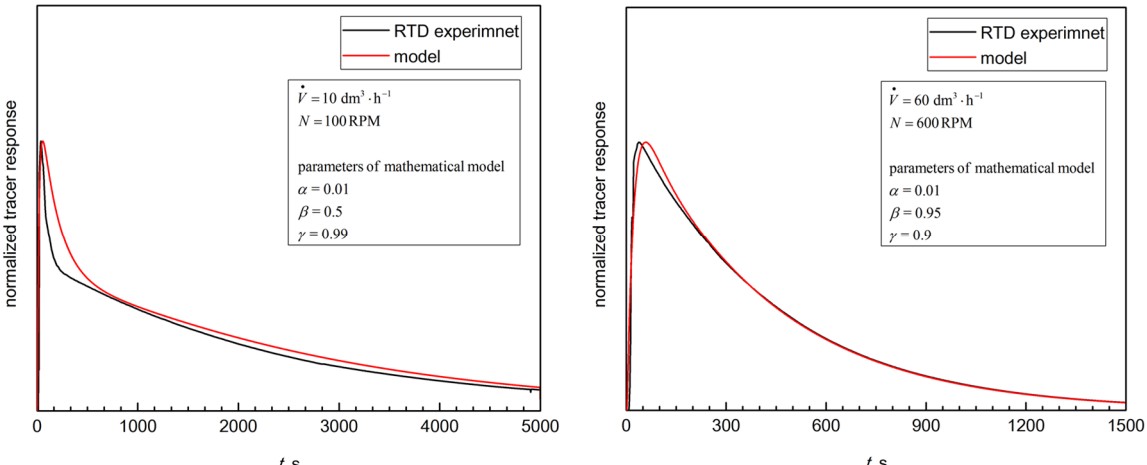

**Figure 7.** The typical examples of the experimental data fit by using the proposed mathematical model (Equation (2)).

The proposed approach may be applied to describe the hydrodynamics in a biological reactor. The proposed compartmental model provides data convergent with RTD measurements. The compartmental structure is based on hydrodynamic data generated by CFD models to incorporate the overall flow patterns and velocity distribution. The CFD procedure based on numerical processes that predict the central transport processes can be used to calculate the number of complicated hydraulic occurrences [33]. Moreover, CFD-generated data allow us to obtain explanations characterized by a high level of physical realism. Results established in CFD simulations might be valuable for the mathematical modeling processes. It should be noted that the CFD approach gives numbers of information at a very high computational demand, and it remains an elaborate approach regardless of the improvement in computing performance [34].

In contrast to CFD, the CMs are easy to use and are time efficient. These models are suitable for the description of the non-ideal mixing behavior in bioreactors [35,36]. A standard procedure of characterization of the non-ideal mixing within the bioprocessing apparatus based on using a multi-zonal interpretation that divides the volume into a system of unified mixing zones [37]. The CM enables to consider a bioreactor as a combination of perfectly mixed CSTRs organized in different structures and especially includes dead zones, recirculation, by-pass, and cross-flow phenomena [38]. It should be noted that this combination of CSTRs should mimic the RTD data [39]. The proper choice of a structure of a model describing the bioreactor's non-ideal flow is still difficult. The flow pattern of the model must possess the most important characteristic of that in the tested bioreactor. Therefore, the CFD turbulence analysis may be successfully applied for the development of CM structure [40]. This approach allowed the CFD model's coupling precision and the CM's simplicity and computing speed. Moreover, this type of mathematical modeling may be a powerful tool in predicting the complex

hydrodynamics and biological reactions interaction in stirred-tank bioreactors [34]. The CFD-based CM is also able to obtain the crucial parameters of macromixing in bioreactors [41].

## 4. Conclusions

The development of experimental and computational approaches facilitates the use of the CFD method as a prognostic tool for mixing behavior in a laboratory-scale bioreactor (BioFlo®415 system). The current study's main aim was to present a method that used a CFD model as a base for the development of a compartmental model to describe the hydrodynamics in the tested bioreactor. In conclusion, the current study revealed that the CFD technique allowed selecting a number of compartments, volumes of compartments, and their connections. The proposed mathematical model of bioreactor showed sufficient agreement with the experimental database (RTD measurements). The present study also proved that the proposed procedure could be used to model the mixing behavior of different impeller systems in stirred tanks applied for bioprocessing. Furthermore, it was shown that the proposed CM could be used to predict the hydrodynamic behavior of the tested bioreactor, which facilitates the further inclusion of a bio-kinetic model. It is essential to obtain a mathematical model's quite simple structure that allows implementing the metabolic models' reaction terms.

**Author Contributions:** Conceptualization, A.K., M.K. (Marian Kordas), M.K. (Maciej Konopacki), B.G. and R.R.; methodology, M.K. (Marian Kordas), M.K. (Maciej Konopacki), B.G., M.J.-S. and R.R.; software, M.K. (Maciej Konopacki), R.R.; validation, A.K., M.K. (Marian Kordas), M.K. (Maciej Konopacki), D.M. and K.W.; formal analysis, M.K., M.K., M.K. (Marian Kordas), M.K. (Maciej Konopacki), B.G., M.J.-S. and R.R.; investigation, A.K., M.K. (Marian Kordas), M.K. (Maciej Konopacki), B.G. and M.J.-S.; resources, A.K. and R.R.; data curation, A.K., M.K. (Marian Kordas), M.K. (Maciej Konopacki), B.G; writing—original draft preparation, R.R.; writing—review and editing, B.G.; visualization, A.K., M.K. (Maciej Konopacki); supervision, R.R.; project administration, R.R; funding acquisition, R.R. All authors have read and agreed to the published version of the manuscript.

**Funding:** This research was funded by the National Science Centre (Poland) within the projects grant no. 2018/31/B/ST8/03170 (granted to R.R.).

**Acknowledgments:** 

**Conflicts of Interest:** The authors declare no conflict of interest. The funders had no role in the design of the study; in the collection, analyses, or interpretation of data; in the writing of the manuscript, or in the decision to publish the results.

## Abbreviations

| | |
|---|---|
| $c$ | tracer concentration, $kg_{NaCl} \cdot (kg_{solvent})^{-1}$ |
| $N$ | rotational speed, RPM |
| $\dot{q}$ | mass flow rate, $kg \cdot s^{-1}$ |
| $t$ | time, s |
| V | compartment volume, $m^3$ |
| $\dot{V}$ | volumetric flow rate, $dm^3 \cdot h^{-1}$ |
| $\alpha, \beta, \gamma$ | parameters of mathematical model (see Equation (2)) |
| $\delta$ | Dirac delta function |
| $\varepsilon$ | sum of squares of deviation between measured and predicted values |
| $\tau_0$ | time-lag, s |
| CFD | computational fluid dynamics |
| CM | compartmental model |
| CSTR | continuous stirred-tank reactor |
| RTD | residence time distribution |

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
