# Peer review of "Mathematical Modeling of Hydrodynamics in Bioreactor by Means of CFD-Based Compartment Model"

_processes, doi:10.3390/pr8101301_

Round 1
Reviewer 1 Report
The authors have proposed a "surrogate" model that mimmicks the behaviour of a detailed CFD simulation. This surrogate model is much faster that the CFD simulation and is capable of predicting the RTD curves obtained with a given reactor. As far as I know this has been done multiple times in the literature, and the aproache is not completely new. Nevertheless, this model may be used to model this type of reactor, which may help researchers that are using it.
The paper is well organized and I did not detect any major flaw in the models. The authors, nevertheless, should explain in more detail the type of reference rotating frame used in the CFD simulations. Furthermore, I think it would be important to mention the paper ( Optimization of Reaction Selectivity Using CFD-Based Compartmental Modeling and Surrogate-Based Optimization, https://doi.org/10.3390/pr7010009) which proposes the same methodology for chemical reactors.
Author Response
We wish to thank the Anonymous Referee #1 for his/her constructive comments and insightful suggestions on our paper. They helped us to substantially improve the quality of the manuscript. As suggested, we have added information about CFD and the additional papers.
Reviewer 2 Report
The work is well presented. The introduction can be improved.most of references are outdated. The introduction part can be improved with referring to more recent studies in this field and how this paper helps with their shortcomings.
Author Response
We wish to thank the Anonymous Referee #2 for his/her constructive comments and insightful suggestions on our paper. They helped us to substantially improve the quality of the manuscript. As suggested, we have added additional papers.